# Cancer Pain Management: A Narrative Review of Current Concepts, Strategies, and Techniques

François Mestdagh [1], Arnaud Steyaert [1] and Patricia Lavand'homme [2,*]

1    Department of Anesthesiology and Pain Clinic, Cliniques Universitaires Saint Luc, University Catholic of Louvain, Av Hippocrate 10, B-1200 Brussels, Belgium; francois.mestdagh@saintluc.uclouvain.be (F.M.)

2    Department of Anesthesiology and Acute Postoperative & Transitional Pain Service, Cliniques Universitaires Saint Luc, University Catholic of Louvain, Av Hippocrate 10, B-1200 Brussels, Belgium

*    Correspondence: patricia.lavandhomme@uclouvain.be; Tel.: +32-2-7641821

**Abstract:** Pain is frequently reported during cancer disease, and it still remains poorly controlled in 40% of patients. Recent developments in oncology have helped to better control pain. Targeted treatments may cure cancer disease and significantly increase survival. Therefore, a novel population of patients (cancer survivors) has emerged, also enduring chronic pain (27.6% moderate to severe pain). The present review discusses the different options currently available to manage pain in (former) cancer patients in light of progress made in the last decade. Major progress in the field includes the recent development of a chronic cancer pain taxonomy now included in the International Classification of Diseases (ICD-11) and the update of the WHO analgesic ladder. Until recently, cancer pain management has mostly relied on pharmacotherapy, with opioids being considered as the mainstay. The opioids crisis has prompted the reassessment of opioids use in cancer patients and survivors. This review focuses on the current utilization of opioids, the neuropathic pain component often neglected, and the techniques and non-pharmacological strategies available which help to personalize patient treatment. Cancer pain management is now closer to the management of chronic non-cancer pain, i.e., "an integrative and supportive pain care" aiming to improve patient's quality of life.

**Keywords:** cancer pain; cancer survivors; neuropathic pain; WHO analgesic ladder; opioid analgesics; non-pharmacological treatments; integrative pain care





## 1. Introduction

Many people are affected by cancer, and its prevalence is increasing as the population is aging. Pain is a common symptom of cancer diagnosis and rises in prevalence throughout and beyond cancer treatment [1,2]. In a recent systematic review, including studies from 2014 to 2021, the overall prevalence of pain in cancer patients was 44% [2]. Moderate to severe pain was experienced by 31% of the patients (vs. 38% of the patients in a 2016 review). Pain prevalence in advanced metastatic and terminal cancer was 55% (vs. 66% in a 2016 review). Thus, it seems that both pain prevalence and pain intensity have declined in the past decade. Nevertheless, the presence of poorly controlled pain still remains a problem for many cancer patients, as shown in evidence highlighted by a recent systematic review (including papers from 2014 to 2020) [3]. An analgesic treatment inadequate to the intensity of pain was identified in about 40% of cancer patients. Elderly patients who usually present with several comorbidities and in patients from countries with a low–medium economic level where the access to analgesic drugs may be restricted (due to high costs or health policy) are of particular concern. Further, it is worth noting that pain management often remains secondary to other cancer treatments that contribute to pain undertreatment [4].

While pain itself is not immediately life threatening, chronic pain remains one of the most frequent and disabling symptoms of cancer. Chronic pain is always associated

with poorer quality of life due to psychological distress (fatigue, depression) and reduced functioning [1]. Some data indicate that the presence of poorly relieved pain may decrease survival rates in cancer [5,6].

Recent developments in oncology, which facilitate better control of tumour growth and thereby reduce the associated phenomena of inflammation, ischemia, and compression [7], have also contributed to the reduction in cancer pain prevalence and severity, improving the patients' quality of life [2]. Targeted treatments have also increased patients' survival and, for some patients, have led to a disease-free outcome (curative treatment). Consequently, a novel population of patients called "cancer survivors" has emerged. According to a recent systematic review, 47% of cancer survivors report the presence of some chronic pain (moderate to severe pain: 28%) in relation to previous treatments like chemotherapy, radiotherapy, or curative surgery or even in relation to a concomitant chronic pain condition unrelated to cancer or cancer treatment [6,8]. Among the pain conditions reported in cancer survivors in direct relation to long-term cancer treatment, one notes chemotherapy-induced peripheral neuropathy (30% prevalence), aromatase inhibitor-associated musculoskeletal syndrome (up to 50% prevalence) and rheumatic pain associated with checkpoint inhibitors (up to 22% prevalence) [8]. Managing chronic pain in this population requires a different approach than that used for people with a limited prognosis.

The last few years have shown substantial evolution and relevant improvements in chronic cancer pain management. A major progress in the field is the recent development of a chronic cancer pain taxonomy and its inclusion in the International Classification of Diseases (ICD-11), thanks to a collaboration between the WHO (World Health Organization) and IASP (International Association for the Study of Pain) [1]. There was clearly a need for a standardized classification of cancer-related pain, allowing greater visibility of the problem and facilitating its recognition in public policy decisions, particularly in low- and middle-income countries where chronic pain is as prevalent as in high-income countries, but pain management is often inadequate due to both limited resources and the low prioritization of the problem. The general diagnostic code "cancer-related pain" only demands that the pain arose in relation to cancer and lasted/recurred for 3 months [1]. Cancer patients experience at least two different types of pain [1], and a neuropathic component is present in 20.9% to 40% of the patients, associated with higher pain intensity, poorer quality of life, and higher analgesics intake [9]. Correct identification of the nature and the cause of pain is mandatory to achieve optimal pain control in any chronic pain patient, including cancer patients and cancer survivors. Accurate diagnosis and classification may lead to important benefits for patients: tailored treatment, better supportive therapies, and more specialist referrals [1].

Another important improvement in cancer pain management is the recent adaptation of the WHO analgesic ladder, used as a simple and valuable guidance since 1986. Until recently, cancer pain management has mostly relied on pharmacotherapy, with opioids being considered the mainstay. The opioid crisis, which has highlighted the life-threatening side effects of opioids, has prompted the reassessment of opioid use to treat pain (acute pain, chronic non-cancer, and cancer pain). Opioid dependence, abuse, and misuse, problems most feared in chronic non-cancer pain patients, are now scrutinized in cancer patients and cancer survivors [10,11]. Moreover, the interest in non-pharmacologic treatments in pain management is increasing.

Finally, like in other medical specialties, patient-reported outcomes (PROs) are now considered key elements in making appropriate treatment decisions. In the past, cancer patients did not report pain spontaneously [12], but today, health-related quality of life has gained interest among cancer patients, including older ones [13]. There is an important trend towards taking the patient's preferences and symptoms into consideration instead of only basing the treatment choice on patient comorbidities and drug toxicity profile [12].

In summary, cancer pain management is now closer to the management of chronic non-cancer pain and should be considered as "integrative pain care". By definition, integrative care includes the combination of two or more healthcare strategies in a multidisciplinary, interdisciplinary, collaborative, consultative, and coordinated context (definition of IASP

Global Year 2023). Integrative pain care may combine treatment strategies from different areas of alternative medicine, traditional medicine, or both. Integrative pain care also involves supportive cares. Such a therapeutic approach is a better fit for the complexity of the pain experience and promotes individual preference as well as engagement of the person being treated when developing a pain treatment plan. [14]

The present narrative review aims to present and discuss the options currently available to manage pain in (former) cancer patients in light of the progress made in the last decade. Particular attention has been paid to the literature published during the last ten years, in light of recent guidelines from specific societies like ASCO (American Society of Clinical Oncology), ESMO (European Society for Medical Oncology), or EAPC (European Association of Palliative Care). The review focuses on the current utilization of opioid analgesics, including the concerns of opioid adverse effects in the era of the "opioid epidemic", the presence of a neuropathic component too often poorly diagnosed and treated, as well as the techniques and non-pharmacological strategies available which may help to personalize the treatment of the patient. The available therapies and their indications are summarized in Table 1.

## 2. Opioids in Cancer Pain Management: An Update of the Mainstay Approach

Opioids still remain the mainstay of moderate to severe cancer pain treatment. Consequently, skilled use of opioid analgesics is crucial to adequate pain relief, taking into account their potential harm. Several guidelines from different societies (EAPC, ASCO, ESMO, and WHO) have been published and regularly updated [15–17]. Opioid prescribing relies on the WHO three steps ladder, first released in 1986.

Weak opioids (codeine, hydrocodone, tramadol) are recommended to initiate pain relief in opioid-naïve patients when pain is reported as mild to moderate, with no difference regarding the drugs' efficacy [16–18]. Weak opioids are usually combined with non-opioid analgesics like paracetamol/acetaminophen and/or non-steroidal anti-inflammatory drugs (NSAIDs). There is no evidence showing that initiating opioid therapy by using a weak drug (step II) will improve the overall management of cancer pain. A similar observation was made regarding the strong opioids (step III drugs).

Strong opioids (morphine, oxycodone, hydromorphone) are recommended when the pain intensity is moderate to severe. It is mandatory to begin with a low dose and to titrate up to obtain an optimal balance between satisfactory analgesia and tolerated side effects. According to an overview of Cochrane Reviews (9 reviews, 152 RCTs, $n$ = 13,524), more than 90% of the patients engaged in opioid treatment will experience meaningful pain relief from oral morphine or fentanyl patch within 10 to 14 days [19]. The Cochrane Review also pointed out that up to 77% of the patients report at least one opioid side effect (mainly constipation and nausea), and 10–20% of cancer patients under opioid treatment need to change the treatment [19].

In previous recommendations published in 2012, the analgesic efficacy was considered to be similar among oral morphine, oxycodone, and hydromorphone [18]. However, opioid response varies among patients, and the "interchangeability" of four morphine-like opioids has been questioned in an interesting multicentre, randomized, phase IV trial among cancer patients ($n$ = 520) receiving oral morphine, oral oxycodone, transdermal fentanyl, or transdermal buprenorphine for 28 days [20]. The worst and average pain intensities decreased in a similar way among the four treatment groups. A daily dose increase occurred in each group (from 33% in oral morphine to 121% in transdermal fentanyl). Switching to another opioid due to poor efficacy varied from 22% (with oral morphine) to 12% (with oral oxycodone), and discontinuation of treatment varied from 27% (with oral morphine) to 14% for transdermal fentanyl. Drowsiness, constipation, and dry mouth occurred in half of the patients. Opioid side effects did not differ regarding gastrointestinal side effects (although transdermal fentanyl was previously found to cause less constipation than oral opioids). In contrast, central nervous system side effects (myoclonus, confusion, hallucinations) were more prevalent with oral morphine (13% vs. 2% in transdermal fentanyl group) [20]. In

their conclusion, the authors pointed out the high percentage of non-responders (8.9–14.4%) and poor responders (11–15.3%) to treatment, meaning that 22% to 26% of the patients had less than a 30% reduction in pain intensity after 28 days opioid treatment. Even patients with a good response needed frequent adjustments in opioid therapy [20]. Although the trial suffered several limitations, it underlines the difficulty to provide an adequate and stable/sustainable management of chronic pain in cancer patients for various reasons like opioid tolerance or poorly tolerated side effects of opioids, disease rapid progression, or pain component poorly responsive to opioid treatment.

Sparing strong opioids for the WHO step III ladder has long been questioned, and the use of a low dose of a strong opioid as an alternative to a weak opioid has been suggested [18]. Indeed, early studies have reported that more than 50% of patients needed to switch from step II to step III within two weeks of treatment, due to a lack of pain control [21]. Besides a lack of efficacy, some weak opioids demonstrate genetic polymorphisms that cause an unpredictable analgesic effect [22]. Weak opioids are often expensive in low- and middle-income countries. A recent international open-label RCT (*n* = 153) has compared a two-step approach versus the standard three-step approach WHO analgesic ladder [21]. The results showed no difference in time to obtain stable pain control between the control group (paracetamol, weak opioid, i.e., tramadol or codeine up to maximal doses) and the experimental group (paracetamol, strong opioid, i.e., morphine or oxycodone titration). In the control group, 53% of the patients needed to change to a strong opioid due to ineffective analgesia within 6 days of treatment initiation (IQR 4–11 days). Patients under strong opioids experimented no more side effects, but had less nausea, and the costs were lower. The trial provides some evidence that a two-step approach may be considered a valuable alternative option for cancer pain management.

### 3. Optimizing Opioid Utilisation When Pain Remains Poorly Controlled

As aforementioned, up to 26% of patients are non-responders or poor responders to opioids [20]. Several causes explain the phenomenon, including disease progression, negative psychological conditions, the presence of a neuropathic component, and breakthrough pain (BTP) [23]. Moreover, opioid misuse [10] and the possible development of some opioid tolerance and/or hyperalgesia may be questioned [24].

Opioid rotation or switching is a common practice to optimize pain management. Opioid rotation is defined as switching from one opioid drug to another or changing an opioid's administration route (useful when a patient's clinical state impairs the pharmacokinetic properties or metabolism of opioid drugs) [25]. Two recent reviews on the topic, including, respectively, 9 publications [26] and 20 publications [25], concluded that pain control may be achieved, while the frequency of opioid side effects is rarely lessened. Further, no opioid drug could be found to be the best. Finally, equianalgesic tables commonly used are not based on high-level scientific evidence, and very often, the dose of the new opioid needs to be increased above the dose initially calculated, with the exception of rotations to methadone, and the ratio for a given opioid may change over time. It is worth noting here that opioid combinations are currently not recommended as evidence is limited. Some authors [26] recommend using methadone as a second opioid when high doses of the first-line opioid are already prescribed.

Methadone, developed in the 1930s, is a potent synthetic opioid analgesic with a high oral bioavailability (67–95%), lack of active metabolites, long half-life, and low cost [27]. Methadone displays unique analgesic properties as it binds to μ- and δ-opioid receptors, possesses anti-NMDA (N-methyl-D-aspartate) properties, and may affect serotonin and noradrenaline reuptake (activation of central nociceptive inhibitory systems and antidepressant effect). Methadone has two isomers: d-methadone displays antagonist activity at the NMDA receptor, and l-methadone interacts synergistically with morphine at the μ-opioid receptor. Moreover, its continuous administration as a μ-agonist induces much less NMDA overexpression, an expression which is associated with opioid tolerance and hyperalgesia [28]. To date, methadone has been reported to be very effective in opioid

switching, specifically when high doses of opioids are already used. However, the conversion from other opioids to methadone is not as easy as the conversion between standard opioids. Because of its complex pharmacokinetic profile, methadone prescription should be made by experienced professionals, i.e., pain specialists (according to guidelines like EAPC) [29]. When used as first-line opioid in opioid-responsive pain, methadone does not provide superior analgesia to morphine [30]. A recent study assessed the efficacy and adverse effects of methadone used as a first-line therapy in cancer patients that were either receiving low doses of opioids (weak opioids or others, at dose < 60 mg oral morphine equivalent/day) or none (i.e., opioid-naïve patients) [28]. Opioid-naïve patients started methadone at 6 mg/day and other patients at 9 mg/day. In both groups of treatment, methadone provided good analgesia with limited adverse effects and a minimal opioid-induced tolerance (low methadone escalation index). However, in high-level socio-economic countries, methadone is rarely used as a first-line opioid but instead kept to treat complex pain due to neuropathic involvement or tolerance/hyperalgesia development. Besides its prescription as a second-line opioid after switching, as aforementioned, methadone can also be used as a co-analgesic. A low dose of methadone (e.g., 5 mg/day at the start) as an adjunct to other opioids has been reported in the treatment of cancer pain in palliative care patients and seems to be both effective and safe [31,32]. In these reports (*n* = 146 [29]; *n* = 410 [32]), methadone as a co-analgesic allowed a significant reduction in pain intensity in 49% to 94% of the treated patients, with a low incidence of side effects (20% of patients, no severe side effects).

Buprenorphine is not typically used as a first-line analgesic in cancer pain. Buprenorphine is a strong opioid with mixed agonist and antagonist properties [33,34]. It is a semi-synthetic partial μ-opioid receptor and ORL-1 receptor agonist and a ƙ- and δ-opioid receptor antagonist. The drug binds to the μ-opioid receptor with a high affinity and has a slow dissociation that contributes to a long duration of action and milder withdrawal symptoms. Unlike other opioids, buprenorphine does not induce μ-opioid receptor internalization, which contributes to explaining a reduced risk of tolerance phenomenon. Further, the drug demonstrates antihyperalgesic effects that last longer than the analgesic effect and that might be linked to its ƙ-opioid receptor antagonism. For these reasons, buprenorphine has been approved for opioid withdrawal and maintenance treatment of opioid dependence. Further, compared with other opioids, buprenorphine cause little to no immunosuppression at therapeutic analgesic doses [34]. In humans, a ceiling effect is observed for respiratory depression but not for analgesia. The oral bioavailability is low (15%) due to extensive first-pass metabolism in the gastrointestinal mucosa and the liver. Buprenorphine does not accumulate in renal failure and is not removed by haemodialysis, keeping analgesia unaffected under these circumstances. In cancer patients, buprenorphine is usually prescribed as a transdermal formulation in case of opioid switching and when suitable for some patients, e.g., in renal failure and patients with mixed pain including a neuropathic component [35]. It is worth noting that the drug is now recommended as first-line treatment of chronic pain in cancer survivors [7].

Tapentadol demonstrates strong analgesic effects in relation to its dual mechanism of action. The drug belongs to a novel class of analgesics as it binds to the μ-opioid receptor with an affinity 10- to 20-fold lower than morphine or oxycodone but also acts as a central norepinephrine reuptake inhibitor (NRI). Both mechanisms of action are synergistic and contribute to the analgesic potency of the drug [36]. The use of tapentadol in cancer pain management is recent, and the drug is only available as oral tablets. In opioid-naïve patients with moderate to severe pain, tapentadol 25–200 mg twice daily is non-inferior to oxycodone 5–40 mg twice daily as well as non-inferior to morphine at a dose ratio of 2.5:1 [36]. Tapentadol treatment is associated with fewer gastrointestinal and central nervous system side effects. It is interesting to note that switching from tramadol (a drug with a dual mechanism of action involving both weak μ-opioid receptor binding and serotonin-norepinephrine reuptake inhibition) to tapentadol may be associated with improved analgesic efficacy. In contrast to tramadol, tapentadol is safe in patients with hep-

atic impairment. Switching from high doses of a strong μ-opioid agonist to equianalgesic doses of tapentadol is also feasible but may induce features of mild opioid withdrawal. Tapentadol is particularly effective in cancer patients with mixed pain and neuropathic pain (haematological malignancies, bone metastasis, chemotherapy-induced) with >75% response to treatment and neuropathic pain symptoms reduction [37].

Ketamine was synthetized in the early 1960s as a dissociative anaesthetic and potent analgesic [38]. The drug is a racemic mixture with S(+) isomer being 3 to 4 times more potent than the R(-) isomer. Ketamine can be administered by multiple routes such as intravenous, intramuscular, intranasal, etc., but oral and rectal routes display poor bioavailability (17 and 25%, respectively) due to first-pass metabolism. Norketamine is an active metabolite with weak potency. Ketamine interacts with several systems (opioid, nicotinic, muscarinic), but its major mechanism of action relies on NMDA-receptor antagonism in the central nervous system. Further, ketamine is called a "use-dependent" drug, i.e., it blocks NMDA channels only if they have already been open by intense or repeated noxious stimuli. Opioid administration activates NMDA receptors resulting in opioid tolerance and hyperalgesia [24]. The administration of ketamine at sub-anaesthetic doses (low doses: <0.5 mg/kg) provides significant analgesic effects with limited side effects, i.e., psychodysleptic or dysphoric effects [38]. Finally, the rapid and potent antidepressant effects of ketamine have been recently highlighted. For all the aforementioned reasons, ketamine may be a useful adjuvant in the treatment of refractory chronic pain. In contrast to intravenous administration, oral ketamine has a limited utility [39]. While a recent Cochrane review [40] found insufficient evidence to recommend ketamine as an adjunctive therapy in cancer pain, several clinical reports underline the benefit of low-dose ketamine infusion (started at 100 mg/24 h, up to 300 mg/24 h) added to opioid analgesics in palliative care unit [41,42]. In these reports (*n* = 70), ketamine infusion significantly reduced pain intensity in 56 to 74% of the patients, with an acceptable tolerance.

Magnesium deserves a few comments in the field of pain management. Magnesium ions regulate the conduction of NMDA receptor channels in the central nervous system. Hypomagnesemia may occur in advanced cancer disease for various reasons and may be associated with refractory pain episodes [43]. Consequently, in the case of poorly controlled pain despite strong opioid intake, blood magnesium levels should be checked. In experimental studies, magnesium sulphate enhances the effect of analgesics acting as NMDA receptor antagonists like ketamine and methadone [44]. Clinical studies are needed to support these experimental observations.

Lidocaine (a local anaesthetic and anti-arrhythmic agent), parenteral infusion, may be used to relieve refractory complex neuropathic or visceral pain in advanced cancer when other treatments have failed [45]. Several mechanisms of action are involved, which include the blockage of voltage-dependent sodium channels on nerve membranes and anti-inflammatory effects [46]. According to the case reports and observational cohorts published to date, the optimal regimen for parenteral lidocaine administration is unclear [47]. Due to a narrow therapeutic index, and potential neurologic and cardiac toxicities, close monitoring of the patient is recommended.

Finally, cannabis-related medicines (CBM) have increasingly gained attention. The recent legalization of cannabinoid consumption in many countries has increased their interest in pain management [48]. Recent reviews, however, agree on the fact that CBM provide low to no effect on chronic pain, including cancer-related pain. Moreover, CBM are associated with central nervous system, psychiatric, and gastrointestinal side effects (i.e., nausea, dizziness). In addition, the long-term effects of regular long-lasting use of CBM remain poorly known [48–51].

## 4. Adverse Effects and Harms Related to Long-Term Opioids Intake

Besides their analgesic effects mediated into the central nervous system, exogenous opioid analgesics also interact with various body systems where they disturb endogenous opioid functioning. Common side effects of opioids are well known, e.g., nausea and

vomiting, constipation, sedation, dizziness, hallucinations, and respiratory depression. The development of tolerance and, in rare cases, the development of hyperalgesia also may occur [24]. Other side effects like endocrine changes, i.e., androgens deficiency and bone demineralization, remain too often underestimated. More recently, the risk of depression associated with long-term opioid prescription has been questioned [52]. Finally, psychological dependence and opioid use disorders (OUD) have gained interest the last few years in relation to the "opioids crisis". Opioid side effects may affect the quality of life of cancer patients, and long-term opioid use may even affect survival, a question that is currently debated [8]. Both inadequate pain relief and opioid administration negatively impact the patient's immune response (either directly on tumour growth or indirectly on immune cell functioning). Well-designed prospective studies are needed, taking into account that adequate pain relief remains a priority in cancer patients.

It is worth noting that side effects, and specifically harmful side effects of opioids, are actually pointed out in chronic pain patients, including cancer patients, and even more feared in the "cancer survivors" population [8]. Among opioid harms, the risks of opioid use disorders (OUD) have long been perceived as extremely low by healthcare providers in cancer-treated patients. In a recent systematic review (literature review of the last 20 years), OUD prevalence reached 8% (up to 20%) among patients with cancer-related pain [10]. These findings clearly demonstrate that aberrant opioid analgesic behaviours (i.e., chemical coping), misuse, and addiction also occur in cancer patients under chronic opioid treatment. A large definition of "OUD" used in the review and the studies' heterogeneity precluded defining a profile of higher-risk patients. Nevertheless, male patients seem to be at higher risk, and opioid overdoses are more frequent in patients treated for head and neck cancers and myeloma. Opioid prescribing and use among cancer survivors are currently under consideration as it is in non-cancer chronic pain patients [11]. Besides the effects of chronic opioid intake on the immune system functioning, long-term opioid therapy may induce a state called "hyperkatifeia", i.e., a negative emotional state involving malaise, irritability, dysphoria, alexithymia, anxiety, and ultimately, mood depression [52]. To fight these negative feelings, patients increase opioid intake (i.e., negative reinforcement), which may lead to overdose, suicide, and mortality. In a recent retrospective-based cohort study ($n$ = 54,509), among the 6% of preoperative opioid-naïve patients who still used opioid analgesics at 6 months after lung cancer surgery, the authors found a 40% higher risk of 2-year all-causes mortality like cancer recurrence and opioids overdose [53]. Moreover, long-term users of strong opioids were at higher risk of poorer survival than users of less potent opioids (OR 1.92 vs. 1.22). Consequently, close follow-up of chronic opioid prescriptions is recommended in cancer survivors [8]. Using the lowest doses of opioids, opioid tapering as soon as possible, and prescription of specific drugs like buprenorphine for maintenance is strongly recommended [54].

## 5. The Problem of Neuropathic Pain

Between 20 and 40% of cancer patients will experience neuropathic pain, defined by the International Association for the Study of Pain (IASP) as pain caused by a lesion or disease of the somatosensory nervous system [55]. In these patients, the peripheral or central nervous system can be affected either by the tumour itself or by its treatment (surgery, chemotherapy, and radiotherapy) [1]. In this section, we will focus on the former. A particularity of cancer-related neuropathic pain is the frequent joint presence of nociceptive pain secondary to the mass effect of the tumour or its metastases, a situation referred to as mixed pain [1,56]. Neuropathic mechanisms play an important role in the pathophysiology of cancer-induced bone pain and may cause metastatic bone pain refractory to standard pain treatments. The presence of neuropathic features was found in 30.8% (95% CI: 23.6 to 39.1%) of the patients suffering from cancer-induced bone pain [57]. A recent large Korean observational study confirmed that the presence of neuropathic pain in cancer patients was associated with higher pain intensity, higher pain interference in daily life, and lower quality of life [58]. In this cohort, less than half the patients suffering from neuropathic

cancer pain received the recommended adjuvant analgesics [58], highlighting the fact that despite the recent addition of specific codes in the latest International Classification of Diseases (ICD-11), neuropathic cancer pain remains under-treated [1].

Since there are specific treatment options available for neuropathic cancer pain (NCP), it is important that a correct diagnosis is established. The diagnosis of neuropathic cancer pain can be challenging and requires a comprehensive evaluation, including a detailed medical history, physical examination, and possibly diagnostic tests. In the absence of a gold standard for the diagnosis of neuropathic pain, the revised grading system for neuropathic pain remains the most widely used set of assessment criteria [59]. These criteria are (1) a history of a lesion or disease of the somatosensory nervous system and pain in a plausible neuroanatomical distribution, (2) pain associated with sensory signs in the same neuroanatomical distribution, and (3) confirmatory diagnostic tests indicating the presence of a lesion or disease of the somatosensory nervous. According to the number of satisfied criteria, the pain can be classified as possibly, probably, or definitively neuropathic [59]. While not formally validated in NCP patients, this approach has been endorsed by the IASP Cancer Pain Special Interest Group [56]. Neuropathic pain screening questionnaires, such as the Leeds Assessment of Neuropathic Symptoms and Signs (LANSS), Douleur Neuropathique 4 (DN4), and PainDetect, can be valuable tools to assess the likelihood of neuropathic pain in individuals, including those with neuropathic cancer pain [60].

The guidelines on the management of cancer pain from the European Society for Medical Oncology recommend that NCP be treated with a combination of opioids and adjuvants when opioids alone are not sufficient [15].

First-line medications used to manage neuropathic pain include tricyclic antidepressants (TCAs, amitriptyline and nortriptyline), serotonin and norepinephrine reuptake inhibitors (SNRIs, duloxetine and venlafaxine), and anticonvulsant drugs (mainly gabapentin and pregabalin) [61]. While fewer studies have specifically investigated these drugs in NCP patients, a 2016 systematic review and meta-analysis found that adding antidepressants or anticonvulsants to opioids reduces pain intensity more than opioids alone, but it concluded that the limited evidence precluded a recommendation on specific adjuvants in combination pharmacotherapy [62]. Two more recent studies have reached similar conclusions. A randomized controlled trial enrolled 70 patients with NCP poorly controlled by a combination treatment of opioids and pregabalin and randomized them to receive either duloxetine 40 mg or a placebo. Significantly more patients in the duloxetine group reported a pain reduction of $\geq$50% (32 vs. 3%, $p = 0.002$) [63,64]. A retrospective chart review included 43 patients and showed that the combination of duloxetine and methadone resulted in a modest reduction in NCP intensity compared to methadone or duloxetine monotherapy [65]. The authors report a mean reduction of 0.9 (SD 3.0, $p = 0.05$) on the pain subscale of the Edmonton Symptom Assessment System, which corresponds to a small effect size [65]. Taken together, the evidence seems in favour of a positive but rather modest effect of anti-neuropathic medication on the intensity of NCP. Moreover, clinicians should always balance the potential benefits of these medications with their potential side effects. TCAs possess anticholinergic properties and can induce sedation, dry mouth, blurred vision, urinary retention, orthostatic hypotension, and tachycardia. Adverse effects of SNRIs include nausea, headache, dizziness, sweating, and arterial hypertension. Patients on gabapentinoids frequently complain of dizziness or somnolence, but both pregabalin and gabapentin are also associated with more serious adverse events, such as abuse [66] and, when combined with opioids, respiratory depression [67].

For localized neuropathic pain, topical treatments are another recommended option [61]. Unfortunately, direct evidence from patients suffering from NCP is scarce. Lidocaine patches have been reported to improve pain intensity in a small observational trial that included 20 patients from a palliative care outpatient clinic. While five patients stopped the treatment due to lack of effect, the mean pain intensity was reduced at the 30-day follow-up (5.2 $\pm$ 1.32 vs. 3 $\pm$ 1.37, $p < 0.05$) [68]. Capsaicin is a potent TRPV1 agonist, whose topical application at high concentrations induces a reversible disappearance

of epidermal free nerve endings and, in some patients, an improvement in neuropathic pain symptoms. Consequently, topical application of 8% high-concentration capsaicin is recommended in the treatment of localized neuropathic pain of various aetiologies [61] and has been shown to provide significant pain relief in chemotherapy-induced peripheral neuropathic pain in a recent systematic review [69]. However, its use for the management of NCP caused by direct tumour involvement has not been documented. Finally, Botulinum Toxin type A, a potent neurotoxin useful to treat focal muscle hyperactivity, also displays analgesic effects independent of its action on muscle tone but in relation to the modulation of neurogenic inflammation, a mechanism possibly involved in some peripheral neuropathic pain conditions. Indeed, several RCTs have shown that subcutaneous injections of Botulinum Toxin into the painful area may be effective, with a rapid onset of action (one week) and long-lasting analgesic effect (3 months) [70,71]. None of the published studies, however, included patients with NCP.

Since patients with NCP often present with mixed pain, opioids are commonly prescribed in combination with first-line anti-neuropathic pain drugs [56]. Indeed, the guidelines on the management of cancer pain from the European Society for Medical Oncology recommend that NCP be treated with a combination of opioids and adjuvants when opioids alone are not sufficient [15]. There is no conclusive evidence in the literature about the superiority of one opioid over another in the treatment of NCP, but some molecules with a dual mode of action might achieve better results in this indication. Tramadol is a weak opioid agonist that also inhibits serotonin and noradrenaline reuptake. It is recommended as a second-line treatment for neuropathic pain [61]. Its use in NCP is supported by a small placebo-controlled randomized trial [72]. Patients receiving tramadol reported an improvement in pain intensity ($-57\%$ vs. $-39\%$, $p < 0.001$), but also in Karnofsky scores, sleep quality, and activities of daily living. Adverse effects were also more common in the tramadol group (67% vs. 22%, $p = 0.007$) [72]. The recently introduced opioid tapentadol acts both as a MOR agonist and selective noradrenaline reuptake inhibitor [73]. Kress et al. randomized nearly 500 patients with moderate to severe chronic malignant tumour-related pain—three-quarters of which reported NCP—to receive either tapentadol, morphine sulphate, or a placebo. Both treatment arms were superior to the placebo, and tapentadol was non-inferior to morphine. Gastrointestinal side effects were less frequent in the tapentadol group [74]. More recently, a retrospective cohort study compared the evolution of pain intensity in 127 patients suffering from NCP who started treatment with tapentadol, methadone, oxycodone, fentanyl, or hydromorphone. The reduction in pain intensity was more pronounced with tapentadol, but this was statistically significant only when compared with oxycodone (1.12, 95% CI 0.75–1.49 vs. 0.33, 95% CI 0.09–0.58) [37]. Finally, switching from another opioid to methadone—a MOR agonist and N-methyl-D-aspartate (NMDA) antagonist—has been shown to improve pain scores in patients with NCP in a recently published prospective cohort pilot study. Allodynia and the pressure/squeezing sensations were the most markedly reduced [75]. Methadone's complex pharmacokinetic profile—most notably its long half-life—and its variable conversion ratio when switching from another opioid complicate its use in clinical practice [29].

## 6. The Problem of Breakthrough Pain

The definition of breakthrough pain (BTP) is a transient exacerbation of pain, with a rapid onset, which occurs while baseline pain is controlled and the patient receives a stable background analgesic regimen [76]. Its incidence is estimated to be between 40 and 80% and is particularly elevated in advanced disease. BTP may occur without a precipitating event or may be triggered by an identifiable event like movement-induced bone pain in metastatic disease or swallow-induced oropharyngeal pain in mucositis, or it may be caused by a diagnostic or therapeutic intervention. The phenomenon presents in different ways in each individual, and may even change its presentation in the same individual during the course of the disease [77]. Since BTP presentation and characteristics are changing, it is challenging to treat, and a personalized and regularly adapted management is required.

First, it is recommended to avoid factors which provoke the pain episodes and to anticipate BTP development with pre-emptive administration of analgesics (e.g., rescue dose of oral morphine before a painful diagnostic or therapeutic intervention). In addition, BTP is usually better treated by the intake of rapid-onset opioids, i.e., lipophilic opioids such as fentanyl or sufentanil administered by transmucosal (sub-lingual, intra-nasal, oral) route [76]. Two important points deserve attention. The first is the role of efficient control of baseline pain. A recent study ($n = 126$) has assessed the prevalence of BTP in patients receiving low doses of opioids to treat background pain [78]. In this population (low background pain intensity, daily dose of oral morphine < 60 mg), BTP prevalence reached 70%. The results also demonstrated that better optimization of background analgesia, though apparently acceptable, may limit the number of BTP episodes. Second, the dose of opioids to be used to relieve BTP remains controversial: either using a dose that is proportional to background analgesia (usually 5 to 20% of the daily dose) or using the minimum available dose to be titrated until the effective dose is reached [77]. Further large studies are needed to answer the question.

## 7. More Personalized Treatment: Potential Benefits of Non-Invasive and Invasive Techniques in Pain Management

The WHO analgesic ladder has been used worldwide since its presentation in 1986, but this approach does not provide adequate pain relief in up to 30% of patients [79]. Updated versions of the WHO ladder have thus been proposed [79–81]. With the new version, integrative medicine can be implemented at every step. A 4th step is proposed, which includes interventional treatments in the WHO ladder. Finally, a "step up, step down" approach allows treatments to be adapted to the type of pain. The conventional slow upward pathway is used for chronic cancer pain, while a faster downward pathway directly starts with the higher steps in the acute phase of pain. When pain is controlled, the clinician can "step down" the ladder to less potent analgesics (Figures 1 and 2). However, the application of the fourth step in the WHO ladder should be considered before refractory pain appears and even before the application of step 3 in some patients who could benefit from more personalized therapeutic strategies [82]. The European Society for Medical Oncology (ESMO) recommends adopting an integrative approach that includes primary antitumour treatments, interventional analgesic therapy, and a variety of non-invasive techniques [15]. While those treatments have a growing body of evidence for cancer pain, studies considering the benefit of such an approach in breakthrough cancer pain are still lacking [15]. Radiotherapy, hormonotherapy, chemotherapy, and surgery can effectively relieve pain in certain cancer patients [82]. Other more specific strategies, somehow more invasive, need to be considered on a "case by case" basis. The presence of a multidisciplinary team is mandatory to optimally manage refractory pain [82].

Photobiomodulation (PBM), i.e., "low-level laser therapy (LLLT)", a non-invasive technique that utilizes wavelengths of light between 650 and 1000 nm to deliver low irradiance to the target tissue, has gained interest over the last two decades [83]. The main effect is the reduction in local inflammation. PBM is particularly useful to alleviate pain conditions secondary to the administration of cancer treatments like mucositis and radiodermatitis [83]. Based on a systematic review, PBM applied in lymphedema after breast cancer therapy might provide short-term relief and might help to reduce limb swelling [84].

Neuromodulation consists of "the alteration of nerve activity through targeted delivery of a stimulus, such as electrical stimulation or chemical agents, to specific neurological sites in the body" [85]. This can be achieved by spinal cord stimulation, neuraxial drug delivery system, peripheral nerve stimulation, or peripheral nerve field stimulation [82].

Deep brain stimulation, repetitive transcranial magnetic stimulation, transcranial direct current stimulation, or motor cortex stimulation are in the field of research [82].

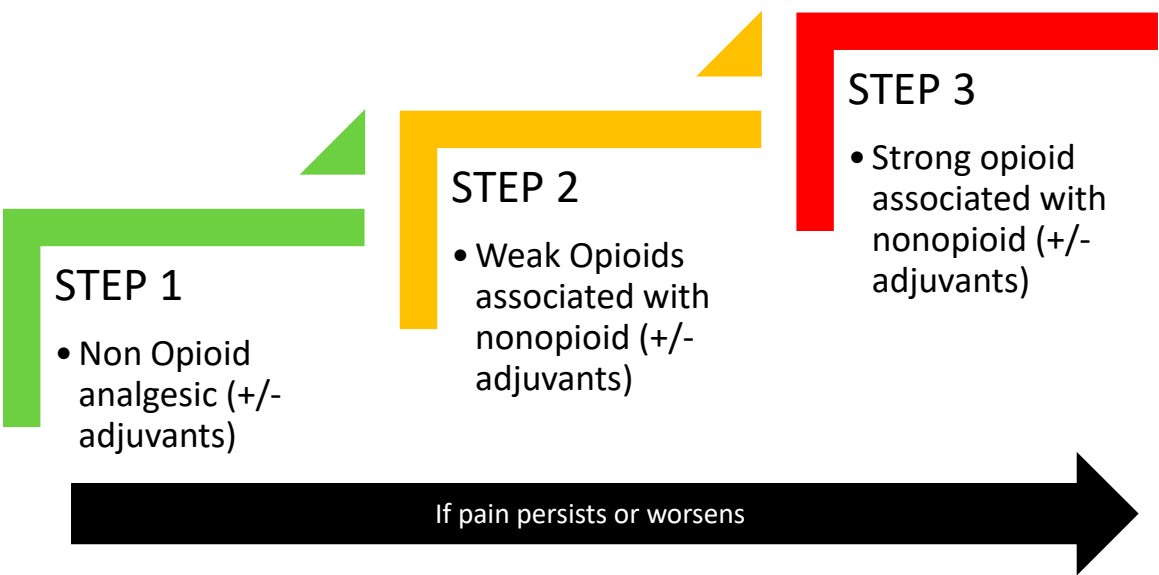

**Figure 1.** The WHO analgesic ladder for treating cancer pain. Adapted from the World Health Organisation.

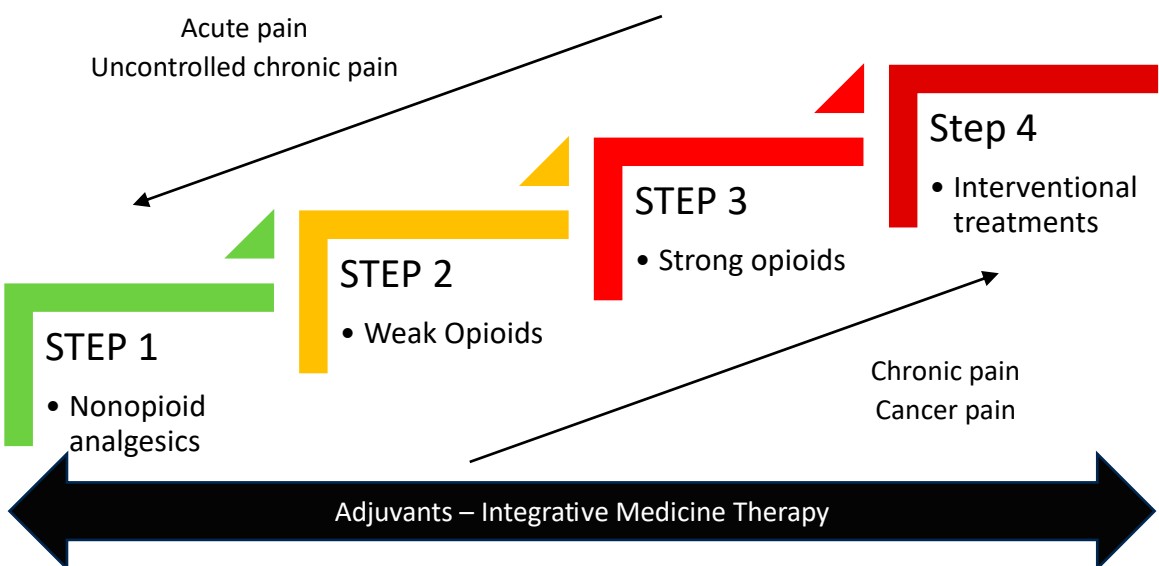

**Figure 2.** New adaptation of the analgesic ladder. This adapted version of the ladder includes interventional treatments and provides a bidirectional approach, depending on the pain and its response to treatments.

Spinal Cord Stimulation (SCS) and dorsal root ganglion stimulation can provide pain relief in neuropathic pain, but studies dedicated to cancer pain are missing. Data are promising, with pain relief of at least 50%, but only case reports have been published up to now [82,86,87].

Neuraxial drug delivery is an option for some patients, i.e., refractory pain, opioids intolerance, widespread bone metastases, and specific locations like pancreatic cancer. Through an intrathecal (IT) catheter, drugs are infused directly near the spinal dorsal horn, bypassing the blood–brain barrier [88,89]. As for now, only ziconotide and opioids have FDA approval and have been proven to be effective and safe by this route. These drugs can be used in association with local anaesthetics, baclofen, clonidine, or ketamine (low to moderate evidence exists for those latter drugs) [82]. The use of the IT route is associated with better pain management and quality of life (QoL), a reduction in systemic

opioid needs (ranging from 300 to 700 mg·d$^{-1}$), and fewer systemic side effects due to the lower doses of opioids used [89–91]. Some authors note that it could also increase patient survival [15,82,91]. The use of an intrathecal drug delivery system (IDDS) is associated with a reduction in health care utilization for patients [90]. Complications relating to the technique (i.e., pump failure, implantation surgery, programming) are rare. Side effects relating to the drugs administered can occur, depending on the dosage. Morphine may induce similar side effects as the systemic route, though less frequent, and the development of granuloma at the catheter tip has been reported. Ziconotide is associated with dizziness, nausea, and confusion [89]. Before implantation, clinicians should consider the diagnosis, expected survival, previous use of opioids, location, and type of pain. Thus, implanted IT catheters and pumps should be reserved for patients who have long-term survival expectations (>3 months), while external IT catheters might be considered for the other patients. The catheter implantation should be decided after appropriate consideration and never proposed as a rescue treatment, which could lead to a failure in pain management [88]. A recent systematic review has highlighted the fact that preimplantation opioid consumption is usually high, suggesting that IDDS often remains a last resort option [90]. A multidisciplinary team and a specialized pain centre are mandatory to manage patients with such devices [82]. Regular routine evaluations and multidisciplinary re-assessment are recommended [82]. It is, however, worth noting that recent expert consensus proposes a wider application of intrathecal analgesia in cancer pain treatment, for reasons including to ensure comfort at the end of life [88].

Technics of percutaneous neurolysis include cryoanalgesia, thermal neurotomy, or pulsed radiofrequency, with the duration of effects depending on the lesion of the nerve.

Percutaneous neurolysis can be used for neuropathic refractory pain in patients with a short life expectancy. The block usually lasts for 3–6 months [15,82,92]. A multimodal guidance with combined imagery techniques is required to perform the technique [82]. A spinal neurolytic block with ethanol or phenol allows the infiltration of the dorsal roots. This technique is limited to pain localized to a few dermatomes and requires a highly skilled team. One should be cautious about a low emergence of the Adamkiewicz artery and the occurrence of vasospasm leading to spinal paralysis [15,82].

Stellate ganglia block can be performed for breast, upper limb, or posterior cervical spine cancer pain and seems to provide a reduction in pain in more than 50% of patients [82].

Coeliac plexus or splanchnic neurolytic block are used for abdominal and epigastric cancer pain, but the results are mixed, depending on the tumour location. The dispersion of the neurolytic solution might be unpredictable and even ineffective when local anatomy is modified by the tumour growth [88]. Severe complications like Adamkiewicz artery vasospasm are rare. Vasodilatation induced by the block increases the upper abdomen temperature and the intestinal motility [82,92].

Hypogastric plexus neurolytic block has been studied in pelvic cancer pain of visceral origin. Only a few studies are available on the topic [82] as well as on the effectiveness of ganglion impar block, which is applied to relieve lower rectal and perianal burns [82,92].

The aforementioned neurolysis techniques may provide good pain relief in the short- and mid-term and contribute to reduce the consumption of systemic drugs. They are usually applied in patients with a short life expectancy, and they can be repeated if needed [15,92]. A cancer progression should be suspected when the analgesic effect is of short duration, while some nerve regeneration may explain the recurrence of pain after a longer time period [82].

Cordotomy is a surgical procedure that consists of provoking lesions to the spinothalamic tract, thus blocking the pain pathway. This therapeutic approach should be reserved for patients who have a short-term survival and suffer severe nociceptive or neuropathic pain. The technique should be performed in hospitals offering specialized palliative medicine, oncology, and pain medicine teams [82].

Percutaneous ablation of metastasis can be achieved through either radiofrequency or cryotherapy. This technique is safe and provides significant pain relief for metastatic bone lesions [82].

Vertebral instability and spinal cord compression are usually treated by surgical procedures. Vertebroplasty or kyphoplasty are now the first approaches, as they can be achieved percutaneously and under local anaesthesia. Cement is injected under radioscopic guidance, with good pain relief [82,93,94]. Cement leakage is common, but significant complications are rare [93]. These minimally invasive techniques represent a safe alternative to manage vertebral compression consecutive to fracture [93,94].

Botulinum toxin (BT) is well known for its effects on muscle contracture, with paresis occurring a few days after injection and lasting for up to 3 months. In addition, BT displays analgesic effects through the reduction in the release of substance P, glutamate, and calcitonin gene-related peptide. A central analgesic effect has also been suggested [95–97]. The administration of BT has shown positive effects in the treatment of migraine and peripheral neuropathic pain conditions [96]. Local BT reduced neuropathic pain and muscle spasms when injected in the vicinity of the radiotherapy area or surgical area, and beneficial effects could last for 12 weeks at least [96–98]. Additionally, in vitro and in vivo studies have demonstrated that BT could induce cellular apoptosis and tumour size reduction [96,97,99]. However, some patients might not respond to BT injection, possibly due to the development of some immune resistance (an effect observed in some patients who underwent repeated BT injections, hence cumulative doses) [99]. BT injection induces few side effects and appears safer than potent analgesic drugs [96,99]. Moreover, potential effects on cancer cell lines are also promising and deserve future developments [96].

## 8. Psychological Support and Non-Pharmacological Therapies

Integrative medicine also involves non-pharmacological therapies, which may reinforce the other therapeutic strategies and are mainly dedicated to improving the patient's comfort and quality of life. Mind–body practice is based on the interactions between brain, mind, body, and behaviour. The mind is used to improve physical function and health. Different techniques are available (i.e., meditation, hypnosis, tai chi, biofeedback, etc.), allowing for a personalized approach according to the patient's preference. These techniques have demonstrated benefits for anxiety, depression, fatigue, and emotional wellness [92,100].

Hypnosis induces a modified state of consciousness with an increased response to suggestion. Self-induced hypnosis is effective in managing pain and improving the quality of life of chronic pain patients, being more effective in highly hypnotizable patients. Nevertheless, evidence remains limited, and more studies are needed in the field of chronic cancer pain [100].

Yoga practice improves the quality of life but, up to now, has shown a beneficial effect on pain only in patients presenting with aromatase inhibitor-related joint pain [101,102]. Studies have related some promising effects of Tai Chi and Qi Gong on emotional wellbeing (i.e., better control of anxiety, depression, stress, and then enhanced quality of life). More studies are still required to support these findings [92,100,103].

Mindfulness and meditation are effective for cancer-related symptoms (i.e., anxiety, fatigue, depression). They improve the global patient's quality of life and might reduce pain severity, as reported in a recent systematic review including American and Danish clinical trials [104]. Based on clinical and psychological characteristics, mindfulness might be more effective for some patients [105].

Cognitive behavioural strategies and pain coping are easily accessible techniques with a beneficial impact on pain symptoms, but their use remains poorly studied in cancer patients [105].

Music therapy, either receptive or active (i.e., playing, singing), can reduce pain, emotional distress, and analgesic drugs consumption. Music therapy interventions can also improve quality of life and fatigue, as shown in recent systematic reviews [105–107].

Acupuncture is used worldwide for diverse reasons, including pain and its application is growing in oncologic practice [108–110]. Moreover, recent systematic reviews seem to demonstrate that acupuncture might be beneficial for cancer pain management including cancer pain in palliative care patients, allowing the reduction in some analgesic drugs intake. However, the quality of the evidence remains weak [103,108,111–113]. Considering the safety of the technique and the limited adverse effects, acupuncture is therefore considered by several authors as part of the integrative approach to cancer pain management and recommended by international societies like ASCO [101,110,113].

Massage therapy has shown a beneficial effect on cancer pain, fatigue, and anxiety in recent systematic reviews, but the evidence is limited and weak [102,114,115]. When considered, caution should be warranted, and massages should not be applied on soft tumour tissues or bone metastasis sites [100].

A systematic review of quality measures for palliative care in oncology has shown that psychological, social, and spiritual aspects of patient suffering are often neglected [116]. In the general population, religious and spiritual interventions may have a small beneficial effect on pain, contributing to reduce physical symptoms and to increase the quality of life, especially in patients enduring a chronic condition (i.e., obesity, cancer) [117]. Spiritual and religious interventions have shown moderate effects on the quality of life in cancer patients, with little impact on pain reduction [118,119]. Considering the safety and the acceptance of these interventions, their use in a holistic approach of the patient is certainly beneficial and even recommended by some authors [92,119].

As a future therapeutic approach, virtual reality is gaining interest due to recent technological progress. Its use can improve the patient's overall well-being and reduce anxiety. The evidence for pain management, especially chronic pain, is still inconclusive, and more studies are needed in this field [120–122].

## 9. Conclusions

Pain is frequently reported during cancer disease, and still remains poorly controlled in around 40% of patients. Recent developments in oncology have helped to better control pain. These targeted treatments may cure cancer disease and significantly increase survival. Therefore, a novel population of patients (also called "cancer survivors") has emerged, with some of them enduring chronic pain (28% reported incidence of moderate to severe pain). Pain management in these patients requires different strategies than the treatment of patients with limited life expectancy. Major progress has been made in the last decade, which includes the recent development of a chronic cancer pain taxonomy now part of the International Classification of Diseases (ICD-11), as well as the update of the WHO analgesic ladder. Until recently, cancer pain management mostly relied on pharmacotherapy, with opioids being considered the mainstay. The "opioids" crisis has prompted the reassessment of opioid use, in both cancer patients and cancer survivors. This recent literature review demonstrates that cancer pain management is now closer to the management of chronic non-cancer pain and should be considered as "integrative pain care". Clinicians should switch to dynamic interdisciplinary pain management. Alternative interventional therapies should be available when the primary approach (i.e., traditional WHO ladder approach) has failed. Since cancer pain is multimorphic, optimal management always requires a dynamic evaluation to constantly adapt the therapeutic approach. Highly specialized teams with appropriate technical support are mandatory [82,105]. Further, the different scientific societies strongly recommend using a multimodal approach, i.e., pharmacological, physical, and psychotherapeutic treatments [82,100,101,105] as well as supportive cares to better fit with personalized treatment. The goal should be to improve the patient's quality of life, not to increase the lifespan at the expense of its quality. The latter observation has become a priority among patients' requests.

**Table 1.** Available therapies for cancer pain management.

| Treatment Options | | Indications | References |
|---|---|---|---|
| **Analgesics** | | | |
| **Nonopioid** | Paracetamol, NSAIDs | Every patient if indicated | [15–17] |
| **Weak opioids** | Codeine, hydrocodone, tramadol | Mild-to-moderate pain in association with nonopioid Tramadol: second line for neuropathic pain | [16–18] |
| **Strong opioids** | - First line: Morphine, oxycodone, hydromorphone<br>- Second line: methadone, buprenorphine, tapentadol | Moderate to severe pain | [18–37] |
| **Adjuvants** | | | |
| **Ketamine** | | Refractory chronic pain | [24,38–42] |
| **Magnesium** | | Hypomagnesemia, refractory cancer pain | [43,44] |
| **Tricyclic antidepressants (TCA)** | Amitriptyline, nortriptyline | Neuropathic pain | [61,62] |
| **Serotonin and norepinephrine reuptake inhibitors (SNRI)** | Duloxetine, venlafaxine | Neuropathic pain | [61,62,65] |
| **Anticonvulsant** | Gabapentin, pregabalin | Neuropathic pain | [61,62] |
| **Cannabis-related medicines** | | Chronic pain | [48–51] |
| **Lidocaine infusion** | | Refractory neuropathic painRefractory visceral pain | [45–47] |
| **Topical treatment** | Lidocaine capsaicin | Localized neuropathic pain | [68,69] |
| **Interventional treatments** | | | |
| **Photobiomodulation** | | Localized neuropathic pain | [83,84] |
| **Neuromodulation** | Deep brain stimulation, transcranial magnetic stimulation, transcranial direct current stimulation | Under research | [82] |
| | Spinal cord stimulation, dorsal root ganglion stimulation | Neuropathic pain | [82,86,87] |
| | Neuraxial drug delivery (morphine, ziconotide, local anaesthetic, baclofen, clonidine, ketamine) | Refractory pain, opioids intolerance, widespread bone metastases, specific location (i.e., pancreatic cancer) | [15,82,88–91] |
| | Percutaneous neurolysis (cryoanalgesia, thermal neurotomy, pulsed radiofrequency)<br>- Stellate ganglia<br>- Coeliac or splanchnic plexus<br>- Hypogastric plexus<br>- Ganglion impar | Neuropathic refractory pain<br>- Breast, upper limb, or posterior cervical spine cancer pain<br>- Abdominal and epigastric cancer pain<br>- Pelvic cancer pain<br>- Lower rectal and perianal burns | [15,82,88,92] |

**Table 1.** *Cont.*

| Treatment Options | | Indications | References |
|---|---|---|---|
| **Cordotomy** | | Severe nociceptive or neuropathic pain in short term survival patients | [82] |
| **Percutaneous ablation of metastasis** | Radiofrequency, cryotherapy | Metastatic bone lesions | [82] |
| **Vertebroplasty and kyphoplasty** | | Vertebral instability and spinal cord compression consecutive to fracture | [82,93,94] |
| **Botulinum toxin** | | Neuropathic pain | [70,71,96–99] |
| **Integrative medicine** | | | |
| **Hypnosis** | | Chronic pain | [100] |
| **Yoga Tai Chi and Qi Gong** | | Aromatase inhibitor-related joint pain Chronic pain | [92,100–103] |
| **Mindfulness and meditation** | | Cancer-related symptoms | [104,105] |
| **Cognitive behavioural strategies** | | Chronic pain | [105] |
| **Music therapy** | | Chronic pain | [105] |
| **Acupuncture** | | Chronic pain | [103,108–112] |
| **Massage therapy** | | Chronic pain | [100,102,114,115] |
| **Spiritual and religious interventions** | | Chronic pain | [92,116–119] |
| **Future therapeutic approach** | | | |
| **Virtual reality** | | Under research | [120–122] |

**Author Contributions:** The three authors (F.M., A.S. and P.L.) equally contributed to the conception, the writing, and the revision of the manuscript. All authors have read and agreed to the published version of the manuscript.

**Funding:** This research received no external funding.

**Conflicts of Interest:** The authors declare no conflict of interest.

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
