# Peer review of "Cancer Pain Management: A Narrative Review of Current Concepts, Strategies, and Techniques"

_curroncol, doi:10.3390/curroncol30070500_

Round 1

Reviewer 1 Report

I found this review interesting, although nothing is new. To me it doesn't bring anything more than ESMO cancer pain guidelines.

I think it seriously lacks of oncological and supportive care culture. Pain management is one of the most important supportive care elements and it has to be integrated to a global approach, including the different characteristics of the cancer care pathway. The word "supportive care" is even not cited.

We can't summarize personalized treatment to WHO step 4 technics, or some particular treatments, yet useful like botanic toxin or ketamine, or to some complementary approach. The whole cancer pain management has to be tailored, personalized and exhaustive as shown and described by Lemaire and al. through the concept of multimorphic cancer pain, integrating all the complexity and disruptive factors (Lemaire A, George B, Maindet C, Burnod A, Allano G, Minello C. Opening up disruptive ways of management in cancer pain: the concept of multimorphic pain. Supportive Care in Cancer, May 2019. DOI: 10.1007/s00520-019-04831-z).

This article doesn't reflect the complexity of cancer pain, and totally conceals oncological treatments and comorbidities impacting pain, and it focuses on very classical approaches. For instance: nothing about painful toxicities and new painful entities among survivors (like immunotherapies or oral chemotherapies) etc.

To me major revisions are required if authors want to be read supportive care teams, palliative care teams and mainly: oncology teams

Reviewer 2 Report

This is a very useful review for clinicians, although based on both old and new references. It would suggest the authors make more ciritical comments about the evidence for the efficacy of many methods they describe. Also, the paper should benefit from reorganization of sections (e.g. psychological methods should be described in a separate section). I did not find any update for amelioration of cancer break trough pain, which is an important part of cancer pain. Neither could I find any update for continuous subcutaneous infusion of opioids, particularly in EOL care of cancer pain patients. Are they better than other? Please, find some detailed comments below: 

1. Introduction

- 3.rd line:” In a recent systematic review, from 2014 to 2021,” is an inaccurate sentence: does it mean original articles in review were published between 2014 and 2021 or that the review was published in 2014 to 2021? Please, correct.

- the next sentence also is inappropriate: Moderate to severe pain was experienced by 30.6% of the patients (vs 38% of the patients in 2016) – when was pain expreinced by 30.6 %? In 2014, or in 2021 or according to the review??

- also, I would recommend to express all figures with similar decimals, i.e. “31” instead“30,6”

2. Opioids in cancer pain management

- p 6: the following sentence is difficult to understand and should be rewritten: Switch to an other opioid varied from 22% (morphine) to 12% (oxycodone) and discontinuation of treatment varied from 27% (morphine) to 14.5% for transdermal fentanyl.

- p6. Is something missing (…) in the following sentence? In contrast, central nervous system side effects e.g. myoclonus, confusion, hallucinations…

-pp 8-9: the semi-headings of buprenorphine and tapentadol are witten in italics and may confuse them to strong opioids; they are not considered strong opioids, and the headings should be corrected;

- p. 10: ketamine and magnesium are dealt in the section of opioids, but they are not opioid analgesics; please, correct

- p. 10 Adverse effects and harms related to long-term opioids intake: please avoid the following kind of expression (…): like mood, immune system…

3. The problem of neuropathic feature in cancer pain

- the better heading would be: the problem of neuropathic pain in cancer patients

- p 13: both pregabalin and gabapentin are also associated with more serious adverse events, such as respiratory depression [56] is wrong and should be corrected: it is not the gabapentinoids but the combination of them with opioids which cause respiratory depression! Furthermore, is the abuse risk significant among patients with neuropathic cancer pain??

4. Personalized treatment: is it possible?

- does ganglion impair block have any role in alleviation of cancer pain?

- this section would benefit from a few sentence about scientific evidence for using invasive methods for cancer pain relief; which of the methods are based on solid evidence (e.g. celiac ganglion block) for efficacy and which are based on clinical findings?

- I would separate psychological methods of pain relief from other methods and describe them under in a separate section

The English language should be carefully reviewed.

Reviewer 3 Report

The methadone section needs to be expanded:

1) methadone also agonizes the delta opioid receptor which has interesting and unique properties

2) conversion from other opioids to methadone is not as straightforward as conversion between standard opioids.  This should at least be mentioned.

Botulinum toxin has also been used in post-thoracotomy pain syndrome with some success.  Will leave that to the authors whether it is worth mentioning.

This article is extremely broad in its scope and therefore is short on deep knowledge regarding the real-life mechanics of cancer pain management.  Perhaps reducing the expectations of the reader that, after reading this article, they will be equipped to treat cancer pain themselves, either the title be reconfigured to state that it is an overview and review of the literature, or a paragraph or two on what tools are available to help readers learn how to properly treat cancer pain in the real world would be helpful.

It's fine, minor editing only.

Round 2

Reviewer 1 Report

Thanks a lot for this new version of the manuscript which has really been improved.

Only two minor revisions to me : 

- botulinum toxin should be cited either in the text and in the table as a topic treatment for neuropathic pain (Attal and al.) with capsaicin and lidocaine

- photobiomodulation should be cited either in the text and in the table as a non invasive interventional approach for chemotherapy induced neuropathy and mucositis/radiodermitis pain management (International WALT 2022 consensus guideline - Robijns J, Nair RG, Lodewijckx J, Arany P, Barasch A, Bjordal JM, Bossi P, Chilles A, Corby PM, Epstein JB, Elad S, Fekrazad R, Fregnani ER, Genot MT, Ibarra AMC, Hamblin MR, Heiskanen V, Hu K, Klastersky J, Lalla R, Latifian S, Maiya A, Mebis J, Migliorati CA, Milstein DMJ, Murphy B, Raber-Durlacher JE, Roseboom HJ, Sonis S, Treister N, Zadik Y, Bensadoun RJ. Photobiomodulation therapy in management of cancer therapy-induced side effects: WALT position paper 2022. Front Oncol. 2022 Aug 30;12:927685. doi: 10.3389/fonc.2022.927685. PMID: 36110957; PMCID: PMC9468822. ; International MASCC guidelines : Elad S, Cheng KKF, Lalla RV, Yarom N, Hong C, Logan RM, Bowen J, Gibson R, Saunders DP, Zadik Y, Ariyawardana A, Correa ME, Ranna V, Bossi P; Mucositis Guidelines Leadership Group of the Multinational Association of Supportive Care in Cancer and International Society of Oral Oncology (MASCC/ISOO). MASCC/ISOO clinical practice guidelines for the management of mucositis secondary to cancer therapy. Cancer. 2020 Oct 1;126(19):4423-4431. doi: 10.1002/cncr.33100. Epub 2020 Jul 28. Erratum in: Cancer. 2021 Oct 1;127(19):3700. PMID: 32786044; PMCID: PMC7540329.)

- intraveinous lidocaine infusions should be cited either in the text and in the table for complex neuropathic pain and complex visceral pain

Reviewer 2 Report

The authors have done a nice job with revision. However, the following items should be considered and corrected in the revision:

Comments to the revision:

1) title: Cancer Pain Management (current concepts, strategies and techniques)

- an addition “narrative review” should be done, because this paper does not review solely evidence-based papers but is in reality descriptive and narrative

2) page 10 Ketamine can be administered by multiple routes (intravenous, intramuscular, intranasal…): - - - please, remove “…” and substitute them with either a comprehensive list OR with “as an example” OR “such as”

3) page 20: Paragraph 8. Psychological support and non-pharmacological therapies

- First, let’s say a few words about cannabis-related medicines is bad scientific English; avoid terms “let´s say”

- cannabis-related substances are pharmacological and should NOT be dealt under sub-title which includes non-pharmacological therapies: either omit cannabis-part or remove it to one of the chapters 3 or 5. Maybe the section “neuropathic pain” could include some warning words about cannabis?

English language requires a comprehensive check
